# Distribution and Diversity of Diaptomid Copepods in Freshwater Habitats of Cambodia (Crustacea: Copepoda: Calanoida: Diaptomidae)

**Rachada Chaicharoen** [1] **and Laorsri Sanoamuang** [2,3,*]

1   Department of Science and Mathematics, Faculty of Science and Technology,
    Rajamangala University of Technology Tawan-ok, Si Racha, Chonburi 20110, Thailand
2   International College, Khon Kaen University, Khon Kaen 40002, Thailand
3   Applied Taxonomic Research Center, Faculty of Science, Khon Kaen University, Khon Kaen 40002, Thailand
*   Correspondence: la_orsri@kku.ac.th

**Abstract:** Freshwater diaptomid copepods in Cambodia are poorly studied, and only seven taxa were known previously. From February to October 2007, 255 samples were collected from 237 freshwater sites in nine provinces representing four regions (northwest, Cardamom and Elephant Mountains, Mekong Lowlands, and east) of Cambodia. Diversity, seasonal occurrence, geographical distribution of each species, and a checklist of diaptomid copepods are documented. In this case, 24 species were recorded, including two undescribed species belonging to the genera *Phyllodiaptomus* and *Tropodiaptomus*. In this case, 14 species are recorded for the first time in Cambodia, and *Mongolodiaptomus formosanus* has been recorded for the first time in Southeast Asia. One species appears to be endemic to Cambodia and nine to the lower Mekong River Basin. *Mongolodiaptomus malaindosinensis* was the most common, frequently recorded throughout the four regions (40.5% of the sampled sites), followed by *Vietodiaptomus blachei* (39.2%), *Eodiaptomus phuvongi* (38.8%), *Mongolodiaptomus formosanus* (31.2%), and *Eodiaptomus draconisignivomi* (27.4%). Very rare species such as *Eodiaptomus sanoamuangae*, *Tropodiaptomus vicinus*, *Mongolodiaptomus calcarus*, and *Tropodiaptomus* sp. were only recorded at 0.4% to 0.8% of the sampled sites. The species diversity of Cambodia is most similar to that of Thailand, where 22 species (91.6%) occur both in Cambodia and Thailand, and 12 (50%) of the species have been recorded in both Cambodia and Vietnam.

**Keywords:** Mekong River; *Mongolodiaptomus formosanus*; *Phyllodiaptomus (P.)* sp.; Southeast Asia; Tonlé Sap; *Tropodiaptomus* sp.

## 1. Introduction

Copepods are a group of micro-crustaceans recorded in nearly every kind of freshwater, brackish, and saltwater habitat. Some free-living species are planktonic, inhabiting the limnetic zone; others are benthic, living on the bottom of freshwater bodies; and rare are groundwater species. Planktonic copepods in the family Diaptomidae, order Calanoida, are usually dominant members of the zooplankton and are major food organisms for small fish and other crustaceans in freshwater ecosystems. More than 440 diaptomid species are reported worldwide, and 92 and 94 species have been recorded from the Oriental and Western Palearctic regions, respectively [1,2]. The biodiversity of diaptomid copepods in the lower Mekong River Basin is rather well studied, particularly in Thailand [3–20], where at least 42 species have been recorded. A total of 33 species have been reported in Vietnam [21–25]. However, fewer investigations into this micro-crustacean have been carried out in Laos [26–28], Myanmar [29,30], and Cambodia [28,29,31–35].

In Cambodia, only seven diaptomid species have been previously reported. In 1951, Brehm reported five species, namely: *Allodiaptomus raoi* Kiefer, 1936; *Heliodiaptomus elegans*

Kiefer, 1935; *Diaptomus javanus* Grochmalicki, 1915; *Eodiaptomus blachei* Brehm, 1951; and *Phyllodiaptomus visnu* (Daday, 1906) [31]. According to current taxonomic revision, the following species are now classified into different genera or species: *Diaptomus javanus* is accepted as *Dentodiaptomus javanus* (Grochmalicki, 1915), *E. blachei* is accepted as *Vietodiaptomus blachei* (Brehm, 1951), and *P. visnu* is accepted as *Mongolodiaptomus botulifer* (Kiefer, 1974). Two more species, *Eodiaptomus draconisignivomi* Brehm, 1952, and *Mongolodiaptomus malaindosinensis* (Lai and Fernando, 1978), were described based on specimens collected from the lake Tonle Sap ("sap" in Cambodian, means "lake") by Brehm (1952) [32], and Lai and Fernando (1978) [33], respectively. Recently, Sanoamuang and Watiroyram recorded three more new species in Cambodia in 2018, 2020, and 2021 (*Mongolodiaptomus mekongensis* Sanoamuang and Watiroyram, 2018; *Phyllodiaptomus (P.) roietensis* Sanoamuang and Watiroyram, 2020; and *Dentodiaptomus orientalis* Sanoamuang and Watiroyram, 2021) [28,34,35].

The Kingdom of Cambodia covers an area of 181,035 km$^2$ (Figure 1a). It is located on the mainland of Southeast Asia and is a part of the lower Mekong River Basin, together with Laos, Thailand, and Vietnam. Cambodia's topography consists primarily of the flat, low-lying central plains region. The plain features the basin of the Tonle Sap, the Bassac River plain, and the floodplains of the lowest Mekong River, which knowingly supports a high diversity of aquatic species. The Tonle Sap is the largest natural lake in Southeast Asia and it is connected to the Mekong River by the 147 km long Tonle Sap River [36]. Mountains and highlands are present around the Central Plains region. The Cardamom Mountains are to the west and southwest of the plain, the Elephant Mountains are to the south, and the Dongrak Mountains and the Eastern Highlands are to the north and northeast [37]. The Mekong, the longest river in Southeast Asia (4350 km in length), flows from its source on the Tibetan Plateau in China through Myanmar, Laos, Thailand, Cambodia, and Vietnam via a large delta into the sea. It runs for about 510 km through Cambodia and can be up to 5 km wide in places [36]. The rich sediment deposited during the rainy season when the Mekong River expands, and floods adds to the fertile growing conditions that exist throughout the Upper Mekong Delta. The Tonle Sap, located in western central Cambodia, connects with the Mekong River at Phnom Penh via a 100-km-long natural channel (Figure 1a). During the dry season, when the water level of the Mekong is low, water flows southeast out of the Tonle Sap into the Mekong River. However, during the rainy season, when the level of the Mekong rises, an extraordinary phenomenon takes place. The expanded and swift-moving Mekong River causes the flow of water in the channel linking the Tonle Sap with the Mekong to reverse, forcing water to drain back into the Tonle Sap and, over time, causing the lake to more than double in size. As a result of this unique occurrence, the Tonle Sap is one of the world's richest supplies of freshwater fish [38].

The climate in Cambodia is hot and humid and is controlled by the monsoon winds. The powerful southwest monsoon winds, which start to blow in mid-May and last until early October, bring heavy rainfall and high humidity. The northeast monsoon brings variable cloud cover, intermittent precipitation, and decreased humidity from early November to mid-March. The country has an average annual rainfall of between 1250 and 1875 mm, with the southwestern mountains, the area with the highest rainfall, receiving nearly 5000 mm per year. The average relative humidity is 81%. Temperatures are high throughout the year, ranging from 28 °C in January to 35 °C in April. During our extensive surveys of diaptomid copepods from a wide range of freshwater habitats (237 sites) in nine provinces of Cambodia, we gathered data on the species richness, their relative frequency, seasonal occurrences, and geographical distribution.

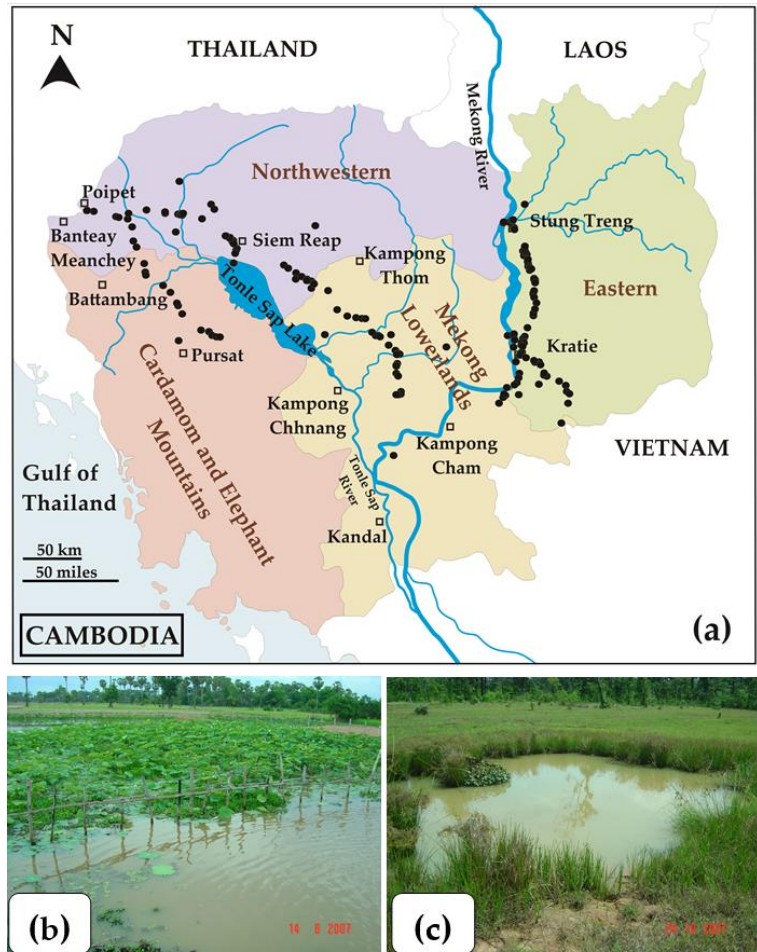

**Figure 1.** Sampling sites in Cambodia: (**a**) a map with 237 sites (black dots) and cities (open black squares) from four geographical regions: the northwestern region, the Cardamom and Elephant Mountains, the Mekong Lowlands, and the eastern region; (**b**) a permanent pond with lotus plants in Kampong Thom Province; and (**c**) a temporary pond in Kratie Province.

## 2. Materials and Methods

### 2.1. Field Work

Diaptomid copepod samples were collected qualitatively from various freshwater habitats in the four regions of Cambodia, covering nine provinces; northwestern region (Banteay Meanchey and Siem Reap), Cardamom and Elephant Mountains (Battambang and Pursat), Mekong Lowlands (Kampong Thom, Kampong Cham, and Kandal), and eastern region (Kratie and Stung Treng) (Figure 1a). A total of 255 samples were collected from 237 sites in three separate periods: during the dry season (February 2007), the early monsoon (June 2007), and the late monsoon (October 2007), using a 60 µm mesh size plankton net. Sampling sites were selected randomly in each season. Although we aimed to collect samples repeatedly at the same sites for three seasons, some sites disappeared by drying in the dry season, flooding in the late monsoon or were replaced by road construction and infrastructure. Thus, three sites were repeatedly collected three times, and 11 sites were repeatedly collected two times.

Samples were taken from 201 permanent waterbodies (53 canals, 4 lakes, 125 permanent ponds (Figure 1b) or swamps, and 19 rivers) and 36 temporary waterbodies (6 rice fields and 30 temporary ponds (Figure 1c)). Samples from each site were preserved both in 4% formaldehyde and 70% ethanol. The habitat types, number of sites, and coordinates (X, Y, Z) of the sampling sites are presented in Table 1, and the sites are shown on the map (Figure 1a).

**Table 1.** Habitat types, number of sites, coordinate ranges, and altitudes of the sampling sites in nine provinces in the four regions of Cambodia. Habitat types are represented as C = canal, L = lake, Pp = permanent pond/swamp, Ri = river, Rf = rice field, and Tp = temporary pond/rain pool. The distribution regions are represented as NW = Northwest, CM = Cardamom and Elephant Mountains, ML = Mekong Lowlands, and E = East.

| Habitat Types | Number of Sites | Latitude (N) | Longitude (E) | Altitude (m) | Region | Province |
|---|---|---|---|---|---|---|
| C, Pp, Rf | 16 | 12°59′–13°34′ | 102°39′–103°33′ | 6–190 | NW | Banteay Meanchey |
| C, L, Pp, Ri, Rf | 31 | 13°03′–13°25′ | 103°23′–104°25′ | 0–82 | NW | Siem Reap |
| C, Pp, Ri, Rf | 20 | 12°40′–13°36′ | 103°06′–105°59′ | 0–11 | CM | Battambang |
| C, Pp, Ri, Rf | 8 | 12°34′–12°42′ | 103°33′–103°49′ | 8–11 | CM | Pursat |
| C, L, Pp, Ri | 37 | 12°12′–13°00′ | 104°33′–105°07′ | 0–52 | ML | Kampong Thom |
| Rf | 1 | 11°53′–11°55′ | 105°29′–105°30′ | 0–16 | ML | Kampong Cham |
| C | 1 | 11°23′–11°25′ | 105°01′–105°03′ | 0–8 | ML | Kandal |
| C, L, Pp, Ri, Rf, Tp | 86 | 11°58′–13°11′ | 106°01′–106°26′ | 5–90 | E | Kratie |
| C, Pp, Ri, Rf, Tp | 37 | 13°17′–13°30′ | 105°59′–106°08′ | 34–92 | E | Strung Treng |
| Total | 237 | | | | | |

At each sampling site, water temperature, pH, and conductivity were measured using a Horiba Water Quality Checker (U-10). Diaptomid copepods were sorted in water and dissected in glycerin under an Olympus stereo microscope. Photographs were taken by a compound microscope (SZX-ILLK 200, Olympus, Tokyo, Japan).

*2.2. Data Analysis*

A one-way analysis of non-parametric statistical method using the Kruskal-Wallis test was utilized to compare the differences in environmental variables between the three seasons. All statistical analyses were performed with SPSS 19.0 software.

The frequency of species occurrence was calculated based on the cumulative number of all collected samples with diaptomid species. The relative occurrences of the diaptomid copepods recorded from Cambodia are categorized according to the occurrence frequencies of the local populations into five categories: (1) common = species were present at more than 30.1% of sampled sites; (2) fairly common = species were present at 10.1–30.0% of sampled sites; (3) uncommon = species were present at 5.1–10.0% of sampled sites; (4) rare = species were present at 1.1–5.0% of sampled sites; and (5) very rare = species were present at less than 1.0% of sampled sites.

## 3. Results

*3.1. Species Diversity*

In this case, 24 species of the family Diaptomidae were present in the 237 sites (255 samples) in the four main regions of Cambodia. A list of the diaptomid copepods with their habitat types, and distribution regions is provided in Table 2. Figure 2 shows pictures of the male leg 5 (P5), which is used to identify the species level of the diaptomid copepods that have been recorded in Cambodia. With six known species, the genus *Mongolodiaptomus* has the most diverse species of the nine genera in the Diaptomidae. This is followed by the genera *Eodiaptomus* and *Phyllodiaptomus* (with four species each), and *Neodiaptomus* (with three species).

**Table 2.** List of the diaptomid copepods recorded during the present study with their habitat types, and distribution regions in the freshwater habitats of Cambodia. Abbreviations of habitat types and distribution regions are given in the caption of Table 1. Species marked with an * are new records for Cambodia.

| No | Species | Habitat Types | | | | | | Distribution Regions | | | | Number of Sites Recorded | % of Sites Recorded |
|---|---|---|---|---|---|---|---|---|---|---|---|---|---|
| | | Permanent Waters | | | | Temporary Waters | | NW | CM | ML | E | | |
| | | C | L | Pp | Ri | Rf | Tp | | | | | | |
| 1 | *Allodiaptomus raoi* Kiefer, 1936 | | ✔ | ✔ | ✔ | | | ✔ | | ✔ | ✔ | 5 | 2.1 |
| 2 | *Dentodiaptomus javanus* (Grochmalicki, 1915) | | | ✔ | | ✔ | ✔ | ✔ | ✔ | | ✔ | 5 | 2.1 |
| 3 | *Dentodiaptomus orientalis* Sanoamuang and Watiroyram, 2021 | | | ✔ | | | ✔ | ✔ | | | ✔ | 5 | 2.1 |
| 4 | *Eodiaptomus draconisignivomi* Brehm, 1952 | ✔ | ✔ | ✔ | ✔ | ✔ | ✔ | ✔ | ✔ | ✔ | ✔ | 65 | 27.4 |
| 5 | *Eodiaptomus phuphanensis* Sanoamuang, 2001 * | ✔ | | ✔ | | ✔ | | ✔ | ✔ | ✔ | ✔ | 21 | 8.9 |
| 6 | *Eodiaptomus phuvongi* Sanoamuang and Sivongxay, 2004 * | ✔ | ✔ | ✔ | ✔ | | ✔ | ✔ | ✔ | ✔ | ✔ | 92 | 38.8 |
| 7 | *Eodiaptomus sanoamuangae* Ranga Reddy and Dumont, 1998 * | | | ✔ | | | | | | | ✔ | 1 | 0.4 |
| 8 | *Heliodiaptomus elegans* Kiefer, 1935 | ✔ | ✔ | ✔ | ✔ | ✔ | | ✔ | ✔ | ✔ | ✔ | 38 | 16.0 |
| 9 | *Heliodiaptomus phuthaiorum* Sanoamuang, 2004 * | ✔ | | | | | | | | ✔ | | 2 | 0.8 |
| 10 | *Mongolodiaptomus botulifer* (Kiefer, 1974) | ✔ | | ✔ | ✔ | | ✔ | ✔ | | ✔ | ✔ | 46 | 19.4 |
| 11 | *Mongolodiaptomus calcarus* (Shen and Tai, 1965) * | | | ✔ | | | | | | ✔ | ✔ | 2 | 0.8 |
| 12 | *Mongolodiaptomus dumonti* Sanoamuang, 2001 * | ✔ | | ✔ | | ✔ | | ✔ | | | ✔ | 3 | 1.3 |
| 13 | *Mongolodiaptomus formosanus* Kiefer, 1937 * | ✔ | ✔ | ✔ | ✔ | ✔ | ✔ | ✔ | ✔ | ✔ | ✔ | 74 | 31.2 |
| 14 | *Mongolodiaptomus malaindosinensis* (Lai and Fernando, 1978) | ✔ | ✔ | ✔ | ✔ | ✔ | ✔ | ✔ | ✔ | ✔ | ✔ | 96 | 40.5 |
| 15 | *Mongolodiaptomus mekongensis* Sanoamuang and Watiroyram, 2018 | ✔ | ✔ | ✔ | ✔ | ✔ | ✔ | ✔ | ✔ | ✔ | ✔ | 37 | 15.6 |
| 16 | *Neodiaptomus laii* Kiefer, 1974 * | ✔ | | ✔ | ✔ | ✔ | | ✔ | ✔ | ✔ | ✔ | 20 | 8.4 |
| 17 | *Neodiaptomus yangtsekiangensis* Mashiko, 1951 * | ✔ | ✔ | ✔ | ✔ | | | ✔ | ✔ | ✔ | ✔ | 12 | 5.1 |
| 18 | *Phyllodiaptomus (Ctenodiaptomus) praedictus* Dumont and Ranga Reddy, 1994 * | ✔ | | | | ✔ | | ✔ | | | | 4 | 1.7 |

**Table 2.** *Cont.*

| No | Species | Habitat Types | | | | | | Distribution Regions | | | | Number of Sites Recorded | % of Sites Recorded |
|---|---|---|---|---|---|---|---|---|---|---|---|---|---|
| | | Permanent Waters | | | | Temporary Waters | | NW | CM | ML | E | | |
| | | C | L | Pp | Ri | Rf | Tp | | | | | | |
| 19 | *Phyllodiaptomus (Phyllodiaptomus) roietensis* Sanoamung and Watiroyrum, 2020 | ✔ | | ✔ | | | | ✔ | ✔ | | | 3 | 1.3 |
| 20 | *Phyllodiaptomus (Phyllodiaptomus)* sp.* | ✔ | | ✔ | | ✔ | ✔ | | ✔ | | ✔ | 9 | 3.8 |
| 21 | *Tropodiaptomus oryzanus* Kiefer, 1937 * | ✔ | | ✔ | ✔ | ✔ | | ✔ | ✔ | ✔ | ✔ | 12 | 5.1 |
| 22 | *Tropodiaptomus vicinus* (Kiefer, 1930) * | | | ✔ | | | | | ✔ | | ✔ | 2 | 0.8 |
| 23 | *Tropodiaptomus* sp. * | | | | | ✔ | | | | ✔ | | 2 | 0.8 |
| 24 | *Vietodiaptomus blachei* (Brehm, 1951) | ✔ | ✔ | ✔ | ✔ | | ✔ | ✔ | ✔ | ✔ | ✔ | 93 | 39.2 |
| | Total | | | | | | | 16 | 16 | 17 | 20 | | |

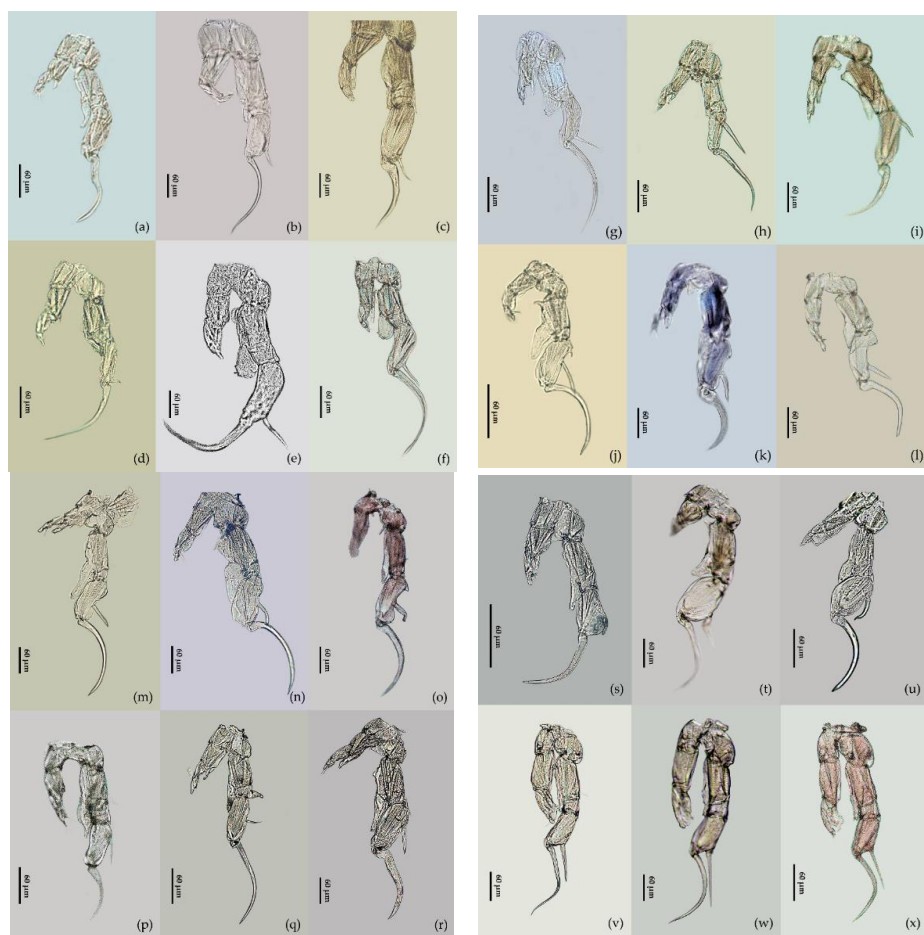

**Figure 2.** Photographs of the male leg 5 (P5) of diaptomid copepods which have been recorded in Cambodia: (**a**) *Allodiaptomus raoi*; (**b**) *Dentodiaptomus javanus*; (**c**) *D. orientalis*; (**d**) *Eodiaptomus draconisignivomi*; (**e**) *E. phuphanensis*; (**f**) *E. phuvongi*; (**g**) *E. sanoamuangae*; (**h**) *Heliodiaptomus elegans*; (**i**) *H. phuthaiorum*; (**j**) *Mongolodiaptomus botulifer*; (**k**) *M. calcarus*; (**l**) *M. dumonti*; (**m**) *M. formosanus*; (**n**) *M. malaindosinensis*; (**o**) *M. mekongensis*; (**p**) *Neodiaptomus laii*; (**q**) *N. yangtsekiangensis*; (**r**) *Vietodiaptomus blachei*; (**s**) *Phyllodiaptomus (C.) praedictus*; (**t**) *P. (P.) roietensis*; (**u**) *P. (P.)* sp.; (**v**) *Tropodiaptomus oryzanus*; (**w**) *T.* sp.; and (**x**) *T. vicinus*.

The following 14 species are new to the fauna of Cambodia: (1) *Eodiaptomus phuphanensis* Sanoamuang, 2001; (2) *E. phuvongi* Sanoamuang and Sivongxay, 2004; (3) *E. sanoamuangae* Ranga Reddy and Dumont, 1998; (4) *Heliodiaptomus phuthaiorum* Sanoamuang, 2004; (5) *Mongolodiaptomus calcarus* (Shen and Tai, 1965); (6) *M. dumonti* Sanoamuang, 2001; (7) *M. formosanus* Kiefer, 1937; (8) *Neodiaptomus laii* Kiefer, 1974; (9) *N. yangtsekiangensis* Mashiko, 1951; (10) *Phyllodiaptomus (C.) praedictus* Dumont and Ranga Reddy, 1994; (11) *P. (P.)* sp.; (12) *Tropodiaptomus oryzanus* Kiefer, 1937; (13) *T. vicinus* (Kiefer, 1930); and (14) *T.* sp. Both *Phyllodiaptomus (P.)* sp. and *Tropodiaptomus* sp. are undescribed taxa, so their descriptions will be published separately.

From our survey, freshwater diaptomid copepods were recorded in 234 out of 237 sites. At the same sampling dates, the species richness of the diaptomid calanoids ranged from 1 to 6 species per site. Most of the sampled sites contained 2 to 4 species per site, but eight sites had six species co-occurring at the same time. *M. malaindosinensis*, *M. botulifer*, and *V. blachei* showed the highest co-occurrences with other species, followed by *E. draconisignivomi*, *H. elegans*, *M. mekongensis*, and *T. oryzanus*.

*3.2. Seasonal Occurrence*

Environmental variables measured at the sampling sites during three seasons are presented in Table 3. The Kruskal-Wallis test indicated significant differences among the three seasons regarding the pH and conductivity of the water ($p < 0.05$). The temperature of the water in the early monsoon season was significantly higher than that in the dry and late monsoon seasons.

**Table 3.** Environmental variables (mean $\pm$ standard deviation, and range) measured at the sampling sites in Cambodia during three seasons. Values followed by the different letter(s) in the same row are significantly different ($p < 0.05$).

| Environmental Variables | Dry Season | Early Monsoon | Late Monsoon |
|---|---|---|---|
| Water temperature (°C) | 30.15 $\pm$ 2.75 [b] (23.8–35.4) | 33.79 $\pm$ 2.58 [a] (22.0–39.2) | 29.45 $\pm$ 2.08 [b] (24.9–33.1) |
| pH | 6.56 $\pm$ 1.24 [c] (3.90–11.10) | 8.15 $\pm$ 0.36 [b] (7.30–9.00) | 10.96 $\pm$ 1.06 [a] (7.00–14.00) |
| Conductivity ($\mu$S cm$^{-1}$) | 160.66 $\pm$ 161.98 [a] (9.1–920.0) | 35.56 $\pm$ 36.76 [c] (11.8–183.3) | 69.02 $\pm$ 67.99 [b] (10.5–328.0) |

Species richness was highest in the early monsoon (21 species), followed by the late monsoon (17 species) and the dry season (16 species). In this case, 12 species were recorded in all the sampling seasons, including *Dentodiaptomus orientalis*, *E. draconisignivomi*, *E. phuphanensis*, *E. phuvongi*, *H. elegans*, *M. botulifer*, *M. formosanus*, *M. malaindosinensis*, *M. mekongensis*, *N. yangtsekiangensis*, *V. blachei*, and *Phyllodiaptomus (P.)* sp. Four species (*H. phuthaiorum*, *P. (C.) praedictus*, *T. vicinus*, and *Tropodiaptomus* sp.) were only present in the early monsoon, while *E. sanoamuangae* and *M. calcarus* were only present in the late monsoon and the dry season, respectively.

The common species, *M. malaindosinensis*, *V. blachei*, *E. phuvongi*, *M. formosanus*, as well as *E. draconisignivomi*, were widely distributed in the four regions, and they were present in all habitat types in all three sampling seasons. During each of the three sampling periods, the dominant species were different. The number of sites where common and fairly common diaptomid species were reported throughout three seasons is presented in Table 4. *E. draconisignivomi* and *M. malaindosinensis* were the dominant species in the dry season, while *V. blachei* and *E. phuvongi* were the dominant taxa in the early monsoon, whereas *M. malaindosinensis* was the dominant species in the late monsoon season.

**Table 4.** The number of sites where common and fairly common diaptomid species were reported throughout three seasons.

| Species | Number of Sites | | | |
| --- | --- | --- | --- | --- |
| | Dry Season | Early Monsoon | Late Monsoon | Total |
| *Mongolodiaptomus malaindosinensis* | 39 | 23 | 34 | 96 |
| *Vietodiaptomus blachei* | 22 | 57 | 14 | 93 |
| *Eodiaptomus phuvongi* | 28 | 46 | 18 | 92 |
| *Mongolodiaptomus formosanus* | 20 | 35 | 19 | 74 |
| *Eodiaptomus draconisignivomi* | 42 | 16 | 7 | 65 |
| *Mongolodiaptomus botulifer* | 15 | 24 | 7 | 46 |
| *Heliodiaptomus elegans* | 20 | 6 | 12 | 38 |
| *Mongolodiaptomus mekongensis* | 1 | 30 | 6 | 37 |

*3.3. Geographical Distribution and Taxonomic Checklist*

The relative occurrences of the 24 diaptomid species recorded from Cambodia are categorized into five groups. *M. malaindosinensis* was the most common, frequently recorded throughout the four regions (40.5% of locations), followed by *Vietodiaptomus blachei* (39.2% of locations), *E. phuvongi* (38.8% of locations), and *M. formosanus* (31.2% of locations). Four fairly common species were *E. draconisignivomi* (27.4% of locations), *M. botulifer* (19.4% of locations), *H. elegans* (16.0% of locations), and *M. mekongensis* (15.6% of locations). The rest includes four uncommon, seven rare, and five very rare species.

The geographical distributions of the 24 diaptomid copepods recorded within Cambodia are presented in Figure 3. With 20 species, the eastern region has the highest diversity, followed by the Mekong Lowlands (17 species), the northwest (16 species), and the Cardamom and Elephant Mountains (16 species). Information about the distributions, seasonal occurrences, and/or taxonomic status of the species recorded is given in alphabetical order by the genera.

*Allodiaptomus raoi* (Figure 2a) is rare in Cambodia and was present in 2.1% of locations in lakes, ponds, or swamps, and rivers in the northwest, Mekong Lowlands, and eastern regions (Figure 3a). This species has been recorded in the dry and late monsoon seasons. It was first described in South India [39], and it has also been reported in Thailand [19], Vietnam [25], and China [40].

*Dentodiaptomus javanus* (Figure 2b) is rare in Cambodia and was present in 2.1% of locations in both temporary habitats (rice fields and temporary ponds) and permanent ponds or swamps. It is distributed in the Cardamom and Elephant Mountains, Mekong Lowlands, and eastern regions (Figure 3b), and has been recorded in the dry and early monsoon seasons. *D. javanus* was first described in Indonesia [41], and it has also been reported in Thailand [19], Vietnam [25], and southern China [40].

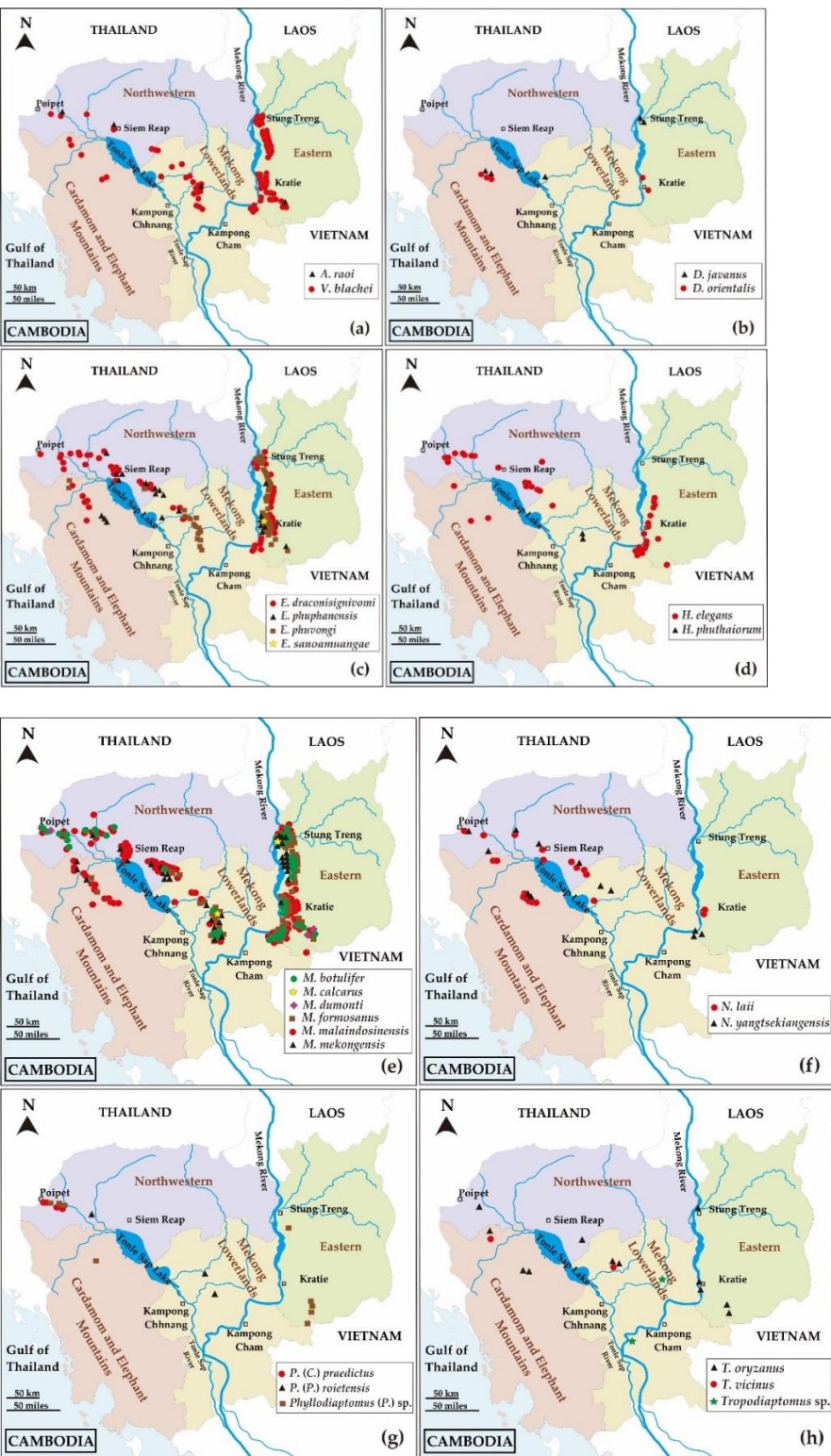

**Figure 3.** Distributions of freshwater diaptomid species in Cambodia during three seasons in 2007: (**a**) *Allodiaptomus* and *Vietodiaptomus*, (**b**) *Dentodiaptomus*, (**c**) *Eodiaptomus*, (**d**) *Heliodiaptomus*, (**e**) *Mongolodiaptomus*, (**f**) *Neodiaptomus*, (**g**) *Phyllodiaptomus*, and (**h**) *Tropodiaptomus*.

*Dentodiaptomus orientalis* (Figure 2c) is rare in Cambodia and was present in 2.1% of locations in both permanent and temporary waterbodies in the Cardamom and Elephant Mountains and eastern regions (Figure 3b) in every season throughout the year. This species

was recently described based on specimens from Thailand and Cambodia in 2021 [35]. It is probably an endemic species of the lower Mekong Basin.

*Eodiaptomus draconisignivomi* (Figure 2d) is common in Cambodia and was present in 27.4% of locations in every kind of habitat in all the four regions (Figure 3c) in every season throughout the year. It was originally described in the neighborhood of the Mekong River, Cambodia, by Brehm in 1952 [32]. This species is also known from Thailand [9,19], Laos [42], and Vietnam [25]. It is a widely distributed diaptomid in the lower Mekong Basin.

*Eodiaptomus phuphanensis* (Figure 2e) is uncommon in Cambodia and was present in 8.9% of locations. It inhabits canals, permanent ponds, and rice fields in the four regions (Figure 3c) in every season throughout the year. This species was first described using specimens from Thailand [10,19], and has also been recorded in Laos [42]. It appears to be an endemic species of the lower Mekong Basin.

*Eodiaptomus phuvongi* (Figure 2f) is the most common species in its genus in Cambodia. It was present in 38.8% of locations in both permanent and temporary waterbodies in all the four regions (Figure 3c) in every season throughout the year. It was first described based on specimens collected from Thailand [19] and Laos [26]. It appears to be an endemic species of the lower Mekong Basin.

*Eodiaptomus sanoamuangae* (Figure 2g) is very rare in Cambodia and was recorded in 0.4% of locations in a permanent pond in the eastern region of the country (Figure 3c) during the late monsoon season. It was first recorded in a roadside canal in Khon Kaen province, northeast Thailand [5]. *E. sanoamuangae* has also been recorded in the Yangtze River and its tributaries in central China [40].

*Heliodiaptomus elegans* (Figure 2h) is fairly common in Cambodia and was present in 16.0% of locations in both permanent and temporary waterbodies in all the four regions (Figure 3d) in every season throughout the year. This species was first described based on specimens from Myanmar [29] and was re-described from samples collected from Thailand by Ranga Reddy and Dumont (1999) [43]. To date, it has been reported from Bangladesh, Vietnam [25], China [40], Thailand [19], and Cambodia.

*Heliodiaptomus phuthaiorum* (Figure 2i) is very rare in Cambodia and was present in 0.8% of locations in canals in the Mekong Lowlands (Figure 3d) in the early monsoon season. It was first recorded in nine sites in the vicinity of the Song Khram River, a tributary of the Mekong River in northeast Thailand [13]. It is probably an endemic species of the lower Mekong Basin.

*Mongolodiaptomus botulifer* (Figure 2j) is fairly common in Cambodia and was present in 19.4% of locations in almost every kind of habitat, except lakes and rice fields, in every region of the country, except the Cardamom and Elephant Mountains (Figure 3e), in every season throughout the year. It has also been recorded in Malaysia, Singapore, Laos, Thailand [19], and Vietnam [25].

*Mongolodiaptomus calcarus* (Figure 2k) is very rare in Cambodia and was present in 0.8% of locations in two permanent ponds in the Mekong Lowlands and eastern regions (Figure 3e) in the dry season. It has also been recorded in Western Java, Indonesia, Malaysia, Singapore, Thailand [19], Vietnam [25], and southern China [40]. Some Malaysian and Singaporean specimens of this species have been mixed up with Indonesian *Neodiaptomus mephistopheles* Brehm, 1933. The correct identification for this species is *M. calcarus* [6,19].

*Mongolodiaptomus dumonti* (Figure 2l) is rare in Cambodia and was present in 1.3% of locations in three waterbodies (a canal, a pond, and a rice field) in the northwestern and eastern regions (Figure 3e) in the early and late seasons. It was described based on specimens recorded in Thailand [12]. This species has been reported so far only in Thailand and Cambodia.

*Mongolodiaptomus formosanus* (Figure 2m) is common in Cambodia and was present in 31.2% of locations in every type of habitat in all the four regions (Figure 3e) in every season throughout the year. This species was formerly believed to be endemic to Taiwan [28], and this is the first time it has been recorded in Southeast Asia.

*Mongolodiaptomus malaindosinensis* (Figure 2n), the most common species of diaptomid in Cambodia, was present in 40.5% of locations in every type of habitat in all the four regions (Figure 3e) in every season throughout the year. This species has also been reported from Singapore, Malaysia [44], Thailand [19], and Vietnam [25].

*Mongolodiaptomus mekongensis* (Figure 2o) is fairly common in Cambodia. It was present in 15.6% of locations in every type of habitat in all the four regions (Figure 3e) in every season throughout the year. This species was discovered recently in the floodplain of the lower Mekong River Basin [28]. Thus far, this species has been recorded in Thailand [19], Vietnam [25], Laos, and Cambodia [28].

*Neodiaptomus laii* (Figure 2p) is uncommon in Cambodia and was present in 8.4% of locations in canals, ponds, swamps, rivers, and rice fields in all the four regions (Figure 3f) in the early and late seasons. This species has also been recorded in Malasia, Singapore, Laos, and Thailand [19].

*Neodiaptomus yangtsekiangensis* (Figure 2q) is uncommon in Cambodia and was present in 5.1% of locations in permanent waterbodies (canals, lakes, ponds, swamps, and rivers) in the four regions (Figure 3f) in every season throughout the year. It has also been reported in Laos, Vietnam [25], Thailand [19], and southern China [40].

*Phyllodiaptomus* (*Ctenodiaptomus*) *praedictus* (Figure 2s) is rare in Cambodia and was present in 1.7% of locations in canals and rice fields in the northwestern region (Figure 3g) in the early monsoon season. This species is a member of the subgenus *Ctenodiaptomus*, which was created by Dumont, Ranga Reddy and Sanoamuang (1996) [4], by having a serrated hyaline fan on the inner margin of the second exopodal segment (Exp-2) of the male left P5. It has also been recorded in Thailand and Laos [19]. A subspecies of *P. (C.) praedictus sulawesensis* Alekseev and Vaillant, 2013, has been reported in Indonesia [45].

*Phyllodiaptomus* (*Phyllodiaptomus*) *roietensis* (Figure 2t) is rare in Cambodia and was present in 1.3% of locations in canals and permanent ponds only in the northwest and Cardamom-Elephant Mountains regions (Figure 3g) in the dry and early monsoon seasons. This species was described in 2020 by Sanoamuang and Watiroyram [34] using specimens from Thailand and Cambodia. It belongs to the subgenus *Phyllodiaptomus*, which is characterized by the Exp-2 of the male left P5 bears a field of spinules.

*Phyllodiaptomus* (*Phyllodiaptomus*) sp. (Figure 2u) is rare in Cambodia and was present in 3.8% of locations in canals, permanent ponds, rice fields, and temporary ponds in the Cardamom and Elephant Mountains and eastern regions (Figure 3g) in every season throughout the year. This undescribed diaptomid is morphologically similar to *P. (P.) christineae* from Thailand. It has also been recorded in Thailand [19].

*Tropodiaptomus oryzanus* (Figure 2v) is uncommon in Cambodia and was recorded in 5.1% of locations in canals, permanent ponds, swamps, rivers, and rice fields in the four regions (Figure 3h) in the early and late monsoon seasons. This species has been recorded in Taiwan, China, Korea, Japan, Thailand [19], and Vietnam [25].

*Tropodiaptomus vicinus* (Figure 2x) is a very rare species in Cambodia and was present in 0.8% of locations in two permanent ponds in the Cardamom and Elephant Mountains (Figure 3h) in the early monsoon season. This species has also been recorded in Indonesia, India, the Philippines, Malaysia, Singapore, Thailand [19], and Vietnam [25].

*Tropodiaptomus* sp. (Figure 2w) is very rare in Cambodia and was recorded in 0.8% of locations in two pools in the rice fields in the Mekong Lowlands (Figure 3h) in the early monsoon season. Morphologically, it is similar to *T. oryzanus*. A description of this species will be published separately. This species is potentially endemic to Cambodia.

*Vietodiaptomus blachei* (Figure 2r) is common in Cambodia and was present in 39.2% of locations in all types of habitats in all four regions (Figure 3a) in every season throughout the year. This species has also been recorded from Laos, Malaysia, Singapore, Thailand [19], Vietnam [25], and Indonesia.

The 24 species recorded from Cambodia can be classified according to their zoogeographical distribution into six groups (Table 5). Only one species (*Tropodiaptomus* sp.) is potentially endemic to Cambodia. Nine species have a more restricted distribution, occur-

ring in the lower Mekong Basin, while the other five species are widely distributed in the countries in the lower Mekong Basin plus the other Southeast Asian countries. Six species are widely distributed in Southeast Asia and East Asia. Two species, *Allodiaptomus raoi* and *Heliodiaptomus elegans*, are the most widely distributed across the lower Mekong Basin, Southeast, East and South Asia. Lastly, *Tropodiaptomus vicinus* is widely distributed across the Southeast and South Asia.

**Table 5.** Geographical distribution of diaptomid copepods so far recorded in Cambodia. The lower Mekong River Basin region consists of Thailand, Myanmar, Laos, Cambodia, and Vietnam. The Southeast Asian region consists of Thailand, Myanmar, Laos, Cambodia, Vietnam, Malaysia, Singapore, the Philippines, Indonesia, and Brunei. The East Asian region consists of China, Japan, North Korea, South Korea, Mongolia, and Taiwan. The South Asian region consists of India, Sri Lanka, Afghanistan, Bangladesh, Bhutan, Nepal, and Pakistan.

| Distribution Regions | Number of Species | List of Species |
|---|---|---|
| Cambodia only (Potentially endemic species) | 1 | *Tropodiaptomus* sp. |
| Lower Mekong River Basin | 9 | *Dentodiaptomus orientalis, Eodiaptomus draconisignivomi, E. phuphanensis, E. phuvongi, Heliodiaptomus phuthaiorum, Mongolodiaptomus dumonti, M. mekongensis, Phyllodiaptomus (P.) roietensis, Phyllodiaptomus (P.) sp.* |
| Southeast Asia | 5 | *M. botulifer, M. malaindosinensis, Neodiaptomus laii, P. (C.) praedictus, Vietodiaptomus blachei* |
| Southeast Asia and East Asia | 6 | *D. javanus, E. sanoamuangae, M. calcarus, M. formosanus, N. yangtsekiangensis, Tropodiaptomus oryzanus* |
| Southeast Asia and South Asia | 1 | *T. vicinus* |
| Southeast Asia, East and South Asia | 2 | *Allodiaptomus raoi, H. elegans* |
| Total | 24 | |

Note: *Eodiaptomus wolterecki* was recorded by Brehm (1951), but it was not found in this study, so it is not included in this table.

## 4. Discussion

Our study's findings bring the number of freshwater diaptomid copepod species recorded in Cambodia up to date from seven [31,32] to 24 species (Table 2). Although *Eodiaptomus wolterecki* was recorded in Cambodia by Brehm in 1951 [31], it was not recorded in this study. The number of diaptomid species recorded is higher than in most Southeast Asian countries, which have between 8 and 19 species, but lower than in Thailand (42 species) [19] and Vietnam (33 species) [25]. The lower species number of Cambodia is reasonable since it has a smaller country area than Thailand (2.83 times) and Vietnam (1.83 times). In addition, the low research intensity and small number of specialists in Cambodia may contribute to a lower number of species compared to Thailand and Vietnam. The species richness in Cambodia is also higher than that of the Amazon basin, with only 16 recorded species [46]. Nevertheless, this species number is not a final figure because samples from this survey were collected from sites that were easy to access along the main roads. There are many more sites in the remote areas of the country that require our visit.

Among the nine genera of Diaptomidae in Cambodia, *Mongolodiaptomus*, *Eodiaptomus*, and *Phyllodiaptomus* have the most diverse species. These three genera are similar to the most species-rich genera in Thailand [19] and Vietnam [25]. Of the 24 diaptomid species recorded in Cambodia, nine (37.5%) belong to species occurring in the lower Mekong River Basin (Thailand, Cambodia, Laos, and Vietnam) (Table 5). The species diversity of Cambodia is most similar to that of Thailand, where 22 species (91.6%) occur both in Cambodia and Thailand [19], and 12 (50%) of the species are present in both Cambodia and Vietnam [25]. Two species (*H. phuthaiorum* and *M. dumonti*) previously considered endemic to Thailand [19] have also been recorded in Cambodia, so they are now categorized as lower Mekong Basin species. Only one species (*Tropodiaptomus* sp.) is potentially endemic to Cambodia, whereas 10 and seven are potentially endemic to Thailand and Vietnam,

respectively. The possible explanation is that many more samples had been collected in Thailand and Vietnam. It is expected that the more samples collected, the more species will be recorded.

Only two taxa (*M. formosanus* and *Tropodiaptomus* sp.) from this study have not been recorded in Thailand. However, our record of *M. formosanus* requires confirmation since the morphological description of this species is sketchy [47] and the classification of this species is confusing. *M. formosanus* and *M. birulai* (Rylov, 1923) were regarded as two separate species by Ranga Reddy et al. in 2000 [7], but *M. formosanus* from Taiwan was treated as a junior synonym of *M. birulai* by Young et al. in 2013 [48]. Therefore, in-depth analyses of this species' morphological and molecular properties are necessary for further research.

The top three most frequently encountered species in the sampled localities in Cambodia (*M. malaindosinensis*, *V. blachei*, and *E. phuvongi*) are comparable to those present in Thailand (*M. botulifer*, *P. praedictus*, and *M. calcarus*) [19] and Vietnam (*E. draconisignivomi*, *M. malaindosinensis*, and *M. botulifer*) [25]. The co-occurrence of the diaptomids at the same sampling dates in Cambodia ranged from 1–6 species per site. This finding confirms that the co-existence of diaptomids in the lower Mekong Basin is the highest recorded, with 1–7 species in Thailand [19] and 1–5 species in Vietnam [25]. A moderately high co-occurrence of 1–4 diaptomid species was reported in India's Northern Western Ghats [39] and in a tropical karst lake district of Mexico [49]. However, in most waterbodies from other regions, only one species was collected from each waterbody, such as in Malaysia [44], Japan [50], and New Zealand [51]. The reason for having a high species number of diaptomids in each site in Cambodia is probably that they are different in body sizes so that they are able to eat different sizes of algae, which are diverse in the lower Mekong Basin [52].

The species richness of diaptomids in Cambodia was highest in the early monsoon (21 species), followed by the late monsoon (17 species) and the dry season (16 species). The high diaptomid diversity in the early monsoon season may be due to the significant differences in the environmental variables (Table 3). The water temperature in the early monsoon season was higher than that in the other two seasons. The pH of the water in the early monsoon season was close to neutral and was significantly higher than that in the dry season but significantly lower than that in the late monsoon season. The suitable water variables for diaptomids in Cambodia that were recorded in the early monsoon were those with a wider range of water temperatures but a narrower range of pH in comparison to those in the other two seasons.

In contrast, those in southern Vietnam showed little difference between the dry season (10 species) and the rainy season (9 species) [25]. The higher species richness of Cambodia in comparison with southern Vietnam may be due to the fact that Cambodia has a complex system of the Tonle Sap, which is connected to the Mekong River.

Considering the diaptomid diversity among the four regions, the highest diversity of 20 species was recorded in the eastern region, while 16–17 species were reported in the other three regions. This can be explained by the fact that 51.9% of the sampling sites were collected from the eastern region, and this area may contain a high diversity of microhabitats. According to the findings of a study conducted on shallow wetlands in South Korea [53], the physical structure of microhabitats has a significant impact on the distribution of various aquatic animals.

## 5. Conclusions

The current study provides the most comprehensive list of 24 freshwater diaptomid species in Cambodia since Brehm's investigation 70 years ago. Although this is not a final number, there are many more sites, particularly temporary ponds, which are one of the most species-rich ecosystems that await our visit. Cambodia might be regarded as one of the biodiversity hotspots for diaptomid copepods in Southeast Asia. The species composition of the diaptomids of Cambodia is similar to that present in neighboring countries in the lower Mekong River Basin, particularly Thailand and Vietnam. One species appears to be endemic to Cambodia and nine to the lower Mekong Basin. Although *Mongolodiap-*

*tomus formosanus* has been discovered here for the first time in Southeast Asia, additional research on this species would require in-depth examinations of its morphological and molecular characteristics.

**Author Contributions:** R.C. contributed to the literature review, sample collection and identification, analyzing data, figure preparation, and writing the early drafts of the manuscript. L.S. contributed to the study's concept and design, literature review, writing, and editing the final manuscript. All authors have read and agreed to the published version of the manuscript.

**Funding:** This research was supported by a grant (RP65–2–Research Center–002) from Thailand's Research and Graduate Studies at Khon Kaen University.

**Institutional Review Board Statement:** Not applicable.

**Data Availability Statement:** The data presented in this study are available in article.

**Acknowledgments:** The authors want to thank Prapatsorn Dabseepai, Seanghun Meas, Waraporn Mahasap and Santi Watiroyrum for their assistance in the sample collections. Supattra Tiang-nga, Kamonwan Koomput and Nattaporn Plangklang are acknowledged for their technical help in the laboratory. The authors gratefully acknowledge two anonymous reviewers for their critical review of the manuscript. Additionally, Nithikarn Sanoamuang is thanked for linguistic corrections to the manuscript. Since the research was carried out in 2007, before the Thai Law on Animals for Scientific Purposes, A.D. 2015, was made public, there is no ethical approval code for this work.

**Conflicts of Interest:** The authors declare no conflict of interest.

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
