# Peer review of "Distribution and Diversity of Diaptomid Copepods in Freshwater Habitats of Cambodia (Crustacea: Copepoda: Calanoida: Diaptomidae)"

_diversity, doi:10.3390/d14110903_

Round 1
Reviewer 1 Report
Comments on the paper:
Distribution and Diversity of Diaptomid Copepods in Freshwater Habitats of Cambodia (Crustacea: Copepoda: Calanoida: Diaptomidae), submitted by Rachada Chaichareon and Laorsri Sanoamuang.
I read this manuscript on Calanoida from Cambodia (South Eastern Asia) with a great interest. I found it as an important contribution to the general knowledge on biodiversity from that area. From this aspect I highly support an idea that this manuscript to be published. However, during my review I found some details (actually a lot), which should be improved. Most of them are of technical nature, which authors can easily correct or change. I made comments/corrections/suggestions directly on the pdf version.
Authors should uniform geographical nomenclature throughout the text – incl. Upper case and lower case.
There is a detail, which I would like to ask authors to add some more information. It is related to ecological parameters, pH and conductivity (what about temperature?), which could be included in the Table 1 or in a separate table or as “Supplementary material”.

Reviewer 2 Report
Manuscript is well written and informative. It gives background of diaptomid study in one of large country in South East Asia which will be useful for further study in both systematics and ecology of this group. However, there are some issues as the followings that I would like to give comments and suggestions.
1. Page 3, Figures 1b-c.
1) please use the same font for every sub-figures
2) what is figure b) represent for? It would be good to address this in the caption and also mentioned in the text.
2. Page 4, Lines 123-129
Please explain how to categorize each rank with those percent of occurrence. Do these percentages come from the histogram analysis of all data? I am a bit wonder with the common species that are only 30% found.
3. Page 4, Table 2
Initials that explained in table caption are the same as in table 1, therefore you can mention the caption that have written in table 1
4. Page 6, Figures 2a-c
I am not sure for the purpose to show these three figures. Please clarify what are the additional information that the authors would like to explain about these three figures.
5. Page 7, Table 3
Are there no overlap sampling sites among 3 seasons? From data in table 3 leads me understand that each species was found in definitely different sites in three studied seasons because the total column is the sum of the all number from three seasons.
Anyway, it is a bit weird because there should be some sites that can be found some species more than 1 time a year?
6. Figures 3-4 showed the same main caption, I do suggest to combine both into one figure. Moreover, I also suggest to show one genus in one sub figure (so there are 9 sub figures in total). Therefore, we can see the whole picture of each genus distribution in only one time (because now some member of some genus presents in different sub figure).
7. Lines 306, 310 Phyllodiaptomus (Ctenodiaptomus) and Phyllodiaptomus (Phyllodiaptomus) roietensis were mentioned here in sub genus level. In my opinion, it should be mentioned in the same taxonomic level both here and other places in the paper. In this part (3.3) you can add more information in the content on their taxonomic status.
8. Lines 367-368: maybe also better compare with the number of researches, sampling sites among Thailand, Vietnam and Cambodia. It seems there are more researches in Thailand than those two countries.
9. The authors also measured the water quality but there are no information mentioned in the results or discussion part. Actually, such data is useful for the discussion.
10. Lines 394-404. Why are there quite high species number of diaptomids in each site in Cambodia?
11. Lines 405-410. What are the ecological differences in the early monsoon comparatively with other seasons in Cambodia which lead to high diversity in each sampling site and rather big different in diaptomid species number with the other two seasons.
12. Lines 411-414. There may be other reasons for the explanation. For example, high diversity of the microhabitats etc. Please add more discussion and refer to the references.
Round 2
Reviewer 1 Report
All requests and corrections from previous version are included into the new version. I have no additional comments or requests.
Author Response
The authors would like to thank Reviewer 1 very much for your very useful comments and suggestions. We very much appreciated your time consuming and detailed review of the manuscript. We accept and incorporate all the comments and suggestions. The revised texts are highlighted in the blue area.
Point 1: Line 13, “northwest and east” - what this means - regions - do they have geographical names?
In a period between February and October or only in February and October? Could you rephrase?
Response 1: Lines 12-14, “northwest and east” are the regions, please see Figure 1a. They do not have geographical names.
We rephrased the sentence to “From February to October 2007, 255 samples were collected from 237 freshwater sites in nine provinces representing four regions (northwest, Cardamom and Elephant Mountains, Mekong Lowlands, and east) of Cambodia.”
Point 2: “Mekong Lowlands” Is this the same as "lower Mekong River basin"? If "yes" please uniform this geographical term (see line 19).
Response 2: No, they are not the same. The Mekong Lowlands is one of the four regions in Cambodia, whereas the lower Mekong River Basin covers a larger area, including Thailand, Laos, Cambodia, and Vietnam.
Point 3: could you rephrase as: .... on the bottom of freshwater water bodies.
What about groundwater species of Copepoda? Calanoida are very rare in groundwater.
Response 3: Lines 35-36, We rephrased the sentence as suggested.
From “ … on the bottom surface of the freshwater wetlands” to “ … on the bottom of freshwater bodies; and rare are groundwater species.”
Point 4: Is it correct "Lower Mekong River Basin" or "lower Mekong River Basin" - see capital letter of lower case.
Response 4: Lines 41, We corrected “Lower” to “lower” as suggested.
Point 5: It is my suggestion: ... from the lake Tonle Sap - I suppose that Sap in Cambodian language means "lake".
Or just explain here that "sap" in Cambodian, means "lake" and use afterwards only "Tonle Sap"
Response 5: Lines 55, We rephrased the text as suggested, “… collected from the lake Tonle Sap ("sap" in Cambodian, means "lake”) …”. “Tonle Sap” is used afterwards throughout the manuscript.
Point 6: Add here: (Figure 1a)
Response 6: Line 61, We added “(Figure 1a)” in the place as suggested.
Point 7: controlled or driven
Response 7: Line 87, We changed “… is governed by …” to “… is controlled by …”
Point 8: put in mm scale (50x25 = 1250 mm; 75X25 = 1875 mm; 200x25 = 5000 mm)
Response 8: Lines 92-93, We changed the inch scale to the mm scale as suggested.
Point 9: add some information on temperature.
Response 9: Lines 94-95, We added a sentence “Temperatures are high throughout the year, ranging from 28 °C in January to 35 °C in April.” as suggested.
Point 10: "Sampling" or "Field work"
Response 10: Line 99, We changed “Sample Collections” to "Field Work" as suggested.
Point 11: samples
Response 11: Line 100, We changed “Diaptomid copepods were …” to “Diaptomid copepod samples were …”
Point 12: Delete this.
Response 12: Line 105, We deleted “(Figure 1)” as suggested.
Point 13: Were each location sampled three times during different seasons or they were sampled randomly actually by seasons? Explain how did you sample each location - in term of seasons.
Response 13: Lines 107–111, We explained more about the sampling sites in three seasons. “Sampling sites were selected randomly in each season. Although we aimed to collect samples repeatedly at the same sites for three seasons, some sites disappeared by drying in the dry season, flooding in the late monsoon or were replaced by road construction and infrastructure. Thus, three sites were repeatedly collected three times, and 11 sites were repeatedly collected two times.”
Point 14: Something is missing here. Start "Samples were ...." as a new paragraph.
Response 14: Lines 111-112, We deleted the incomplete sentence, "The nine provinces of the studied areas are located in the four".
Point 15: Did you mix formaldehyde and alcohol or some samples were preserved in formaldehyde and some in alcohol. Or ALL samples were preserved in formaldehyde and were later transferred into alcohol during sorting material in a lab?
Response 15: Lines 118-119, We rephrased the sentence to “Samples from each site were preserved both in 4% formaldehyde and 70% alcohol.”
Point 16:..., coordinates (X,Y,Z) of the sampling sites are presented in Table 1, and ....
Response 16: Lines 120-121, We rephrased the sentence to read: “…, coordinates (X, Y, Z) of the sampling sites are presented in Table 1, and ....”
Point 17: Start as a new paragraph. Are data on pH and conductivity available in Supplementary material? If not, could you give a range of both values per season and per region or maybe per habitat? Maybe you can include them into Table 1. If not, then delete this sentence.
Response 17: Line 122, The sentence (starting from “At each sampling site …” is started as a new paragraph. We added a new table (Table 3) in the Results part (line 220) to show the means and ranges of environmental variables measured at the sampling sites in three seasons.
Point 18: Probably they were sorted in water or formaldehyde but observed and dissected in glycerol.
Response 18: Line 123, We rephrased the sentence to read: “… were sorted in water and dissected in glycerin…”.
Point 19: Figure 1a, Explain what means "Northeastern" and "Western" - are this provinces? Do the same in the text.
Please, check the scale on Figure 1a - just above "CAMBODIA" - 50 km is not similar to 500 miles!! (In my opinion should be put 50 miles) Correct the scale on other figures, too.
Add solid line on the right side of the map.
Response 19: Lines 129-133, We rephrased and added more information to the figure caption to read "Figure 1. Sampling sites in Cambodia: (a) a map with 237 sites (black dots) and cities (open black squares) from four geographical regions: the northwestern region, the Cardamom and Elephant Mountains, the Mekong Lowlands, and the eastern region; (b) a permanent pond with lotus plants in Kampong Thom Province; and (c) a temporary pond in Kratie Province." The explanation of the four regions in the text had already been done in lines 98-102. We corrected the scale in Figure 1a and added a solid line on the right side of the map.
Point 20: Figure 1a. What means open black squares? Cities? Explain
Response 20: Line 130, We explained the open black squares in the legend of Figure 1a.
Point 21: The caption of Table 1, Explain northwest and east - province, region?
Response 21: Lines 139-140, We rewrote the sentence in the caption of Table 1. “The distribution regions are represented as NW = the Northwest, CM = the Cardamom and Elephant Mountains, ML = the Mekong Lowlands, and E = the East.”
Point 22: probably 30.1 - see numbers below.
Response 22: Line 151, We changed it to “30.1” as suggested.
Point 23: maybe: "very rare" and correct throughout the text, incl. Tables.
Response 23: Line 154, We corrected “extremely rare” to "very rare" throughout the manuscript.
Point 24: .... genera Eodiaptomus anf Phyllodiaptomus (with four species each), .....
Response 24: Lines 163-164, We rephrased as suggested, “... genera Eodiaptomus and Phyllodiaptomus (with four species each), ....”
Point 25: Table 2, I would suggest to organise the first column as follows: put in the upper line latin name and in the second line author+year - to make table uniform in term of line height
I would suggest to delete the last column from the table (Relative Occurences" and use terms "common ------ rare in Cambodia " in the text after line 216.
Response 25: Line 167, Table 2 is revised as suggested. We deleted the last column and use terms "common --- rare in Cambodia" in the text.
Point 26: When the members of the same genus appear in a row, only for the first time a whole genus name should be expelled, for the next species only the first letter of the genus name should be used,
Response 26: Lines 179-189, We corrected the genus names as suggested.
Point 27: Probably: 40.5 % locations, - the same for the rest of numbers
Response 27: Lines 244-248, We added “% of locations” as suggested.
Point 28: Did you sample each location three times? Or just randomly? Put this information in M&M section.
Response 28: 107–111, We explained more about the sampling sites in three seasons in M&M section (lines 104–108). Please read the explanation in Response 13.
Point 29: I would suggest to include here yours estimation of "Relative occurrence" from Table 3 something like: "Species XY is common / very rare in Cambodia, which was present ... (here add locations). Then you can add information on its distribution in SEA, where the species maybe is not so "common or rare".
Response 29: We rephrased the text as suggested.
Point 30: Maybe this part could be omitted - species composition is evident from the legends. Maybe you can replace it with something like " ... in Cambodia during three seasons in 2007."
Response 30: Lines 395-398, We changed the legend of Figure 3 as suggested. Reviewer 2 also suggested we combine Figures 3 and 4 into one figure and to show one genus in one subfigure. Thus, we rewrote the caption as below.
“Figure 3. Distributions of freshwater diaptomid species in Cambodia during three seasons in 2007: (a) Allodiaptomus and Vietodiaptomus, (b) Dentodiaptomus, (c) Eodiaptomus, (d) Heliodiaptomus, (e) Mongolodiaptomus, (f) Neodiaptomus, (g) Phyllodiaptomus, and (h) Tropodiaptomus.”
Point 31: I would suggest: Geographical distribution of diaptomid copepods so far recorded in Cambodia.
Response 31: Lines 410-411, We changed the legend of Table 5 as suggested.
Point 32: Maybe you can change "found" with "recorded" throughout the text.
Response 32: We changed "found" with "recorded or present" throughout the text.
Point 33: From where?
Response 33: Line 424, We added “in Cambodia” in the sentence.
Point 34: I don't understand this sentence. Could you rephrase it? Or delete it.
Response 34: Lines 425-426, The sentence “Since E. wolterecki has never been recorded in the lower Mekong Basin before, the record of this species needs reconfirmation.” is deleted.
Point 35: But it depends also on intensity of research effort:):) and number of specialists:(:)
Response 35: Lines 431-433, We added the sentence “In addition, the low research intensity and small number of specialists in Cambodia may contribute to a lower number of species compared to Thailand and Vietnam.” in the text.

Reviewer 2 Report
I have read the revised version and authors’ response letter and found that the revisions that you made to the manuscript are very effective in addressing the remaining concerns. Therefore, my decision is to accept the manuscript for publication in the present form.
Author Response
The authors would like to thank Reviewer 2 very much for your very useful comments and suggestions. We very much appreciated your time consuming and detailed review of the manuscript. We accept and incorporate all the comments and suggestions. The revised texts are highlighted in the green area.
Point 1: Page 3, Figures 1b-c.
1) please use the same font for every sub-figures
2) what is figure b) represent for? It would be good to address this in the caption and also mentioned in the text.
Response 1: 1) We used the same font for every sub-figures.
2) We rephrased the caption of Figure 1 to read “Figure 1. Sampling sites in Cambodia: (a) a map with 237 sites (black dots) and cities (open black squares) from four geographical regions: the northwestern region, the Cardamom and Elephant Mountains, the Mekong Lowlands, and the eastern region; (b) a permanent pond with lotus plants in Kampong Thom Province; and (c) a temporary pond in Kratie Province.”
We also mentioned it in the text (please see lines 113 and 115).
Point 2: Page 4, Lines 123-129
Please explain how to categorize each rank with those percent of occurrence. Do these percentages come from the histogram analysis of all data? I am a bit wonder with the common species that are only 30% found.
Response 2: We ranked the occurrence frequencies of the 24 Cambodian species from the highest (40.5%) to the lowest (0.4%). Please see the occurrence frequencies in the table below. The difference between the highest and the lowest percentage was 40.1%, and the mean value was 11.4%. Thus, it is reasonable to use 40.5−10.0 = 30.5% as a minimum percentage for the common species. Because 14 species or 58.3% of the 24 species have a percentage of locations less than 10.0%, we chose 10.0% as the maximum percentage for the uncommon species.
We added a sentence to explain how to categorize the relative occurrence in the 2.2. Data Analysis (lines 146-147). “The frequency of species occurrence was calculated based on the cumulative number of all collected samples with diaptomid species.”
|
No. |
Species |
% of locations |
Frequency ranges |
Relative occurrences |
|
1 |
M. malaindosinensis |
40.5 |
> 30.1% |
Common = 4 species |
|
2 |
V. blachei |
39.2 |
||
|
3 |
E. phuvongi |
38.8 |
||
|
4 |
M. formosanus |
31.2 |
||
|
5 |
E. draconisignivomi |
27.4 |
10.1−30.0% |
Fairly common = 4 species |
|
6 |
M. botulifer |
19.4 |
||
|
7 |
H. elegans |
16.0 |
||
|
8 |
M. mekongensis |
15.6 |
||
|
9 |
E. phuphanensis |
8.9 |
5.1−10.0% |
Uncommon = 4 species |
|
10 |
N. laii |
8.4 |
||
|
11 |
N. yangtsekiangensis |
5.1 |
||
|
12 |
T. oryzanus |
5.1 |
||
|
13 |
Phyllodiaptomus sp. |
3.8 |
1.1−5.0% |
Rare = 7 species |
|
14 |
A. raoi |
2.1 |
||
|
15 |
D. javanus |
2.1 |
||
|
16 |
D. orientalis |
2.1 |
||
|
17 |
P. praedictus |
1.7 |
||
|
18 |
P. roietensis |
1.3 |
||
|
19 |
M. dumonti |
1.3 |
||
|
20 |
H. phuthaiorum |
0.8 |
< 1.0% |
Very rare = 5 species |
|
21 |
M. calcarus |
0.8 |
||
|
22 |
Tropodiaptomus sp. |
0.8 |
||
|
23 |
T. vicinus |
0.8 |
||
|
24 |
E. sanoamuangae |
0.4 |
Point 3: Page 4, Table 2
Initials that explained in table caption are the same as in table 1, therefore you can mention the caption that have written in table 1
Response 3: We changed the caption of Table 2 as suggested (line 166). “Table 2. List of the diaptomid copepods recorded during the present study with their habitat types and distribution regions in the freshwater habitats of Cambodia. Abbreviations of habitat types and distribution regions are given in the caption of Table 1. Species marked with an * are new records for Cambodia.”
Reviewer 1 suggested to delete the last column in the table (Relative Occurrences) and use terms "common ------ rare in Cambodia " in the text.
Point 4: Page 6, Figures 2a-c
I am not sure for the purpose to show these three figures. Please clarify what are the additional information that the authors would like to explain about these three figures.
Response 4: We provide more information about the showing of these three figures in the figure caption (lines 191-194).
Figure 2. Photographs of the male leg 5 (P5) of diaptomid copepods which have been recorded in Cambodia: (a) Mongolodiaptomus formosanus, a new record for Southeast Asia; (b) Phyllodiaptomus sp., an undescribed species; and (c) M. malaindosinensis, the most common species in freshwater habitats of Cambodia.
Point 5: Page 7, Table 3
Are there no overlap sampling sites among 3 seasons? From data in table 3 leads me understand that each species was found in definitely different sites in three studied seasons because the total column is the sum of the all number from three seasons.
Anyway, it is a bit weird because there should be some sites that can be found some species more than 1 time a year?
Response 5: We explained more about the sampling sites in three seasons (lines 107–111). “Sampling sites were selected randomly in each season. Although we aimed to collect samples repeatedly at the same sites for three seasons, some sites disappeared by drying in the dry season, flooding in the late monsoon or were replaced by road construction and infrastructure. Thus, three sites were repeatedly collected three times, and 11 sites were repeatedly collected two times.”
Point 6: Figures 3-4 showed the same main caption, I do suggest to combine both into one figure. Moreover, I also suggest to show one genus in one sub figure (so there are 9 sub figures in total). Therefore, we can see the whole picture of each genus distribution in only one time (because now some member of some genus presents in different sub figure).
Response 6: We combined Figures 3 and 4 into one figure. We show species of one genus in one sub-figure, but Fig. 3a shows two species that belong to two genera (Allodiaptomus and Vietodiaptomus) because there is only one species in each genus (lines 395-398).
Figure 3. Distributions of freshwater diaptomid species in Cambodia during three seasons in 2007: (a) Allodiaptomus and Vietodiaptomus, (b) Dentodiaptomus, (c) Eodiaptomus, (d) Heliodiaptomus, (e) Mongolodiaptomus, (f) Neodiaptomus, (g) Phyllodiaptomus, and (h) Tropodiaptomus.
Point 7: Lines 306, Phyllodiaptomus (Ctenodiaptomus) and Phyllodiaptomus (Phyllodiaptomus) roietensis were mentioned here in sub genus level. In my opinion, it should be mentioned in the same taxonomic level both here and other places in the paper. In this part (3.3) you can add more information in the content on their taxonomic status.
Response 7: Lines 350-362, We provided information about the two subgenera in the content as suggested. It was also mentioned at the same taxonomic level both here and in other places in the paper.
Point 8. Lines 367-368: maybe also better compare with the number of researches, sampling sites among Thailand, Vietnam and Cambodia. It seems there are more researches in Thailand than those two countries.
Response 8: Lines 431-433, We added the sentence “In addition, the low research intensity and small number of specialists in Cambodia may contribute to a lower number of species compared to Thailand and Vietnam.” in the text.
Point 9. The authors also measured the water quality but there are no information mentioned in the results or discussion part. Actually, such data is useful for the discussion.
Response 9: Lines 204-208, We provided more information about the environmental variables in the Results’ part. “Environmental variables measured at sampling sites during three seasons are presented in Table 3. The Kruskal-Wallis test indicated significant differences among the three seasons regarding the pH and conductivity of the water (p < 0.05). The temperature of the water in the early monsoon season was significantly higher than that in the dry and late monsoon seasons.”
We also added comments in the discussion part (lines 475-482).
Point 10. Lines 394-404. Why are there quite high species number of diaptomids in each site in Cambodia?
Response 10: Lines 469-472, We added a sentence in the text to explain the reason. “The reason for having a high species number of diaptomids in each site in Cambodia is probably that they are different in body sizes so that they are able to eat different sizes of algae, which are diverse in the lower Mekong Basin [52].”
Point 11. Lines 405-410. What are the ecological differences in the early monsoon comparatively with other seasons in Cambodia which lead to high diversity in each sampling site and rather big different in diaptomid species number with the other two seasons.
Response 11: Lines 475-482, We added sentences regarding the ecological differences in the early monsoon comparatively with other seasons in Cambodia. “The high diaptomid diversity in the early monsoon season may be due to the significant differences in the environmental variables (Tables 3). The water temperature in the early monsoon season was higher than that in the other two seasons. The pH of the water in the early monsoon season was close to neutral and was significantly higher than that in the dry season but significantly lower than that in the late monsoon season. The suitable water variables for diaptomids in Cambodia that were recorded in the early monsoon were those with a wider range of water temperatures but a narrower range of pH in comparison to those in the other two seasons.”
Point 12. Lines 411-414. There may be other reasons for the explanation. For example, high diversity of the microhabitats etc. Please add more discussion and refer to the references.
Response 12: Lines 491-494, We added the following phrases in the text, “… and this area may contain a high diversity of microhabitats. According to the findings of a study conducted on shallow wetlands in South Korea [53], the physical structure of microhabitats has a significant impact on the distribution of various aquatic animals.”
